# Chaos Controllability in Non-Identical Complex Fractional Order Chaotic Systems via Active Complex Synchronization Technique

**Mohammad Sajid** [1,*] **, Harindri Chaudhary** [2] **and Santosh Kaushik** [3]

1. Department of Mechanical Engineering, College of Engineering, Qassim University, Buraydah 51452, Saudi Arabia
2. Deshbandhu College, University of Delhi, New Delhi 110019, India
3. Bhagini Nivedita College, University of Delhi, New Delhi 110043, India
* Correspondence: msajd@qu.edu.sa

**Abstract:** In this paper, we primarily investigate the methodology for the hybrid complex projective synchronization (HCPS) scheme in non-identical complex fractional order chaotic systems via an active complex synchronization technique (ACST). Appropriate controllers of a nonlinear type are designed in view of master–slave composition and Lyapunov's stability criterion (LSC). The HCPS is an extended version of the previously designed projective synchronization scheme. In the HCPS scheme, by using a complex scale matrix, the system taken as slave system is asymptotically synchronized with another system taken as the master system. By utilizing a complex scale matrix, the unpredictability and security of communication are increased along with image encryption. An efficient computational method has been employed to validate and visualize the HCPS method's efficacy by performing numerical simulation outcomes in MATLAB (version 2021).

**Keywords:** active control; complex fractional order chaotic system; hybrid complex projective synchronization; Lyapunov's stability criterion; fractional derivative; simulation

**MSC:** 34K23; 34K35; 37B25; 37N35

## 1. Introduction

Chaos may be characterized as extreme randomness or utter confusion. Chaotic dynamics [1] has currently become the most prominent and intriguing field for researchers. Chaotic systems (CSs) are mainly unpredictable and hugely erratic in behavior. A typical innate property of CSs, which Henri Poincare first announced, is the sensitivity dependency on its initial condition, i.e., 2 close-by points of state space would be separated very quickly with the evolution of time. A microscopic variation in the dynamics of a CS will lead to destructive results.

In 1990, Pecora and Carroll [2] evolved an approach for the synchronization in similar chaotic systems. Chaotic synchronization is a phenomenon that concerns with the coupling of two or more chaotic systems possessing different initial conditions to achieve identical dynamics along with synchronization error converges asymptotically to zero with the evolution of time. After further studies, researchers introduced distinct types of synchronization methods, viz., complete [3], anti [4,5], lag [6,7], phase and anti-phase [8], combination [9], multi-switching [10], hybrid [11], projective [12], hybrid projective [13,14], hybrid function projective [15] and hybrid complex projective synchronization (HCPS) [16,17] by using different control methods such as active control [4,17,18], linear and nonlinear feedback technique [19–21], adaptive control [22–24], sliding mode control [25], active and adaptive sliding mode [26–28], robust adaptive sliding mode [9] and back-stepping design [29].

Fractional calculus [30–32], which was announced in the seventeenth century on 30 September 1965 by Guillaume de Leibniz and L'Hopital, provides the theory for derivatives and integrals of arbitrary order, that combine and establish the notion of integer order differentiation and $n$-fold integration. Fractional calculus gives an important tool for depicting memory as well as inherent effects in distinct substances and fractional calculus is majorly applicable in control theory, dielectric polarization, signal processing, robotics, information processing, finance models, viscoelasticity [33], electromagnetic waves [34], chaotic systems [1], mathematical biology [17], delay differential equations [32], etc.

Fractional order chaotic systems synchronization [35,36] is gaining larger attraction and interest because of its applications in encryption, secure communication, and many more. Fractional order models are basically represented through fractional differential equations or pseudo-state space descriptions [37]. There are several fractional order chaotic systems, for instance, Lorenz [38], Lu [39], Chua [40], T, Rossler, Chen's, and Duffing systems. To escalate complexity, researchers have introduced fractional order complex CSs. As compared to integer order complex network, fractional order complex systems enhance a degree of freedom by utilizing fractional order derivative. Fractional order derivatives may be described in various expressions, for example, Riemann–Liouville form, Caputo form and Grunwald–Letnikov form. Generally, the Caputo fractional order derivative [30,31] operator was pronounced for the complex network since its initial conditions data are similar to that of integer order differential equation conditions. Thus, it provides prominent known physical truths. Therefore, Caputo fractional derivative operator is preferred instead of Riemann–Liouville fractional derivative operator. Another important difference between the Riemann–Liouville derivative definition and the Caputo derivative definition is that although the Caputo derivative of the constant is zero, the Riemann–Liouville fractional derivative of the constant is not equal to zero for a finite value of arbitrary real $\alpha$ [31]. Hence, the Caputo fractional derivative operator becomes the most important form in comparison to other fractional derivative forms.

Various types of synchronization schemes which have been discussed above, are being used to synchronize these complex networks. HCPS, which is an extension of projective synchronization, has been developed not long ago. In HCPS, all scaling functions of the vector are dissimilar due to the complexity increases and improves the strength of secure communication. Despite hybrid function projective synchronization where the scaling factors are in the form of functions, in HCPS the scaling factors are complex-valued. In HCPS, the transformation matrix is a square matrix whose elements are complex. This type of transformation matrix plays a vital role in such cases as chaotic secure communication where the state of the drive system is changed by the scaling factors to send to the communication channel and to enhance the security of the effective information signal.

In this paper, we have considered a fractional order complex Lorenz system as a master system and fractional order complex T-system as a slave system. More importantly, generalizing the 3D Lorenz model to a 5D Lorenz model by modifying the existing state real variable to the complex introduced firstly by Fowler et al. [41]. HPCS scheme via active control has been used to investigate synchronization between these two chaotic complex systems. In the considered master–slave configuration, the states of the slave system are evolving over the period of time, which is guided by the active controller, and this active controller is obtained in view of Lyapunov's stability criterion (LSC) by the error dynamics equations using the HCPS scheme. This error dynamics is a result of both master and slave chaotic systems. As a consequence, we have the required error dynamics. Further, this error dynamics has been used in order to show that the derivative of the designed Lyapunov function is a negative definite function. Consequently, the considered error dynamics achieve global and asymptotical stability with the evolution of time. So, the state trajectories of the master system and the slave system behave alike. Numerical simulations have been done in MATLAB to validate and visualize our results in diagrams.

The main attributes of our suggested research work in this manuscript are enumerated as:

- The suggested HCPS methodology considers two dissimilar complex fractional order chaotic systems.
- It designs a robust HCPS strategy-based control input to achieve hybrid complex projective synchronization among considered fractional order complex systems and performs oscillation for synchronization errors with a fast rate of convergence.
- The description of HCPS scheme-based active control inputs is executed in a simplistic manner utilizing LSC and drive-response/master–salve configuration.
- Simulation outcomes depict the efficacy and superiority of the suggested HCPS strategy.

Describing the remainder of this paper as follows: Section 2 recalls some basic preliminary results, which will be utilized in the coming sections. Section 3 formulates the problem of the HPCS strategy. Section 4 deals with illustrating the synchronization phenomena using the HPCS scheme and active control strategy. Section 5 depicts numerical simulation, which is performed in MATLAB toolbox for establishing the efficacy and suitability of our HCPS scheme. Finally, Section 6 presents some important concluding remarks.

## 2. Preliminaries

Although, many definitions of fractional order derivatives are available in the existing literature. We specifically consider Caputo's derivative definition that is given by:

$$_{t_0}D_t^\alpha f(t) = \frac{1}{\Gamma(n-\alpha)} \int_{t_0}^t \frac{f^{(n)}(\tau)}{(t-\tau)^{\alpha-n+1}} d\tau, \quad t > t_0,$$

where $n = min\{k \in \mathbb{N} \mid k > \alpha\}$, $\Gamma$ denotes Gamma function and the operator $_{t_0}D_t^\alpha$ is usually called '$\alpha$' order Caputo differential operation. Throughout, we denote $_{t_0}D_t^\alpha$ by $D^\alpha$.

**Property 1.** *If $f(t)$ is a constant function and of the order $q > 0$, the Caputo fractional-order derivative satisfies the conditions:*

$$D^q f(t) = 0.$$

**Property 2.** *The Caputo derivative satisfies the following linear property:*

$$D^q[c_1 f_1(t) + c_2 f_2(t)] = c_1 D^q f_1(t) + c_2 D^q f_2(t),$$

*where $f_1(t)$ and $f_2(t)$ are functions of t, and $c_1$ and $c_2$ are constants.*

## 3. Problem Formulation

Consider the following fractional order chaotic system described by

$$D^\alpha U(t) = PU(t) + f(U(t)), \tag{1}$$

where $U = [u_1', u_2', ..., u_n']^T \in C^n$ is a state vector. $P \in C^{n \times n}$ is the constant matrix, $f : C^n \to C^n$ is non-linear function. System (1) is supposed as a drive system.

Corresponding to (1) the slave system with control input vector $\sigma(t) \in C^n$ is given as

$$D^\alpha V(t) = QV(t) + g(V(t)) + \sigma(t), \tag{2}$$

where $V = [v_1', v_2', ..., v_n']^T \in C^n$ is a state vector, $Q \in C^{n \times n}$ is the constant matrix, $g : C^n \to C^n$ is non-linear function and $\sigma(t) = [\sigma^1, \sigma^2, ..., \sigma^n] \in C^n$ is the active control function.

The purpose of this article is to design a suitable active control function $\sigma(t)$ such that the controlled slave system (2) asymptotically approaches against the master system (1) and hence we achieve synchronization.

In HCPS, the synchronization error between the master system (1) and the controlled slave system (2) is given as:

$$E(t) = V(t) - \mu U(t), \tag{3}$$

where $E = [e'_1, e'_2, ..., e'_n] \in C^n$ and $\mu = diag(\mu'_1, \mu'_2, ..., \mu'_n)$ is a complex scaling matrix. Particularly, the coupled master–slave system is said to achieve complete synchronization and anti-synchronization if all the coefficients $\mu$ are equal to 1 and $-1$, respectively.

**Definition 1.** *The master system (1) and the controlled slave system (2) are said to achieve HCPS if there exists a control function $\sigma(t) = [\sigma'_1, \sigma'_2, ..., \sigma'_n] \in C^n$ such that*

$$\lim_{t \to \infty} \| E(t) \| = \lim_{t \to \infty} \| V(t) - \mu U(t) \| = 0, \tag{4}$$

*where $\| . \|$ is the Euclidean norm of a vector.*

The error dynamics is obtained as:

$$\begin{aligned} D^\alpha E(t) &= QV(t) - P\mu U(t) + g(V(t)) - \mu f(U(t)) + \sigma(t) \\ &= (Q + P)E(t) + G(U(t), V(t)) + \sigma(t), \end{aligned} \tag{5}$$

where $G(U(t), V(t)) = g(V(t)) - \mu f(U(t)) + \mu QU(t) - PV(t)$.

To stabilize the error system, we design appropriate control functions $\sigma(t)$ using active control methodology. The control input $\sigma(t)$ is defined as:

$$\sigma(t) = K(t) - G(U(t), V(t)). \tag{6}$$

By using (6), the system (5) reduces to

$$D^\alpha E(t) = K(t) + (Q + P)E(t), \tag{7}$$

then the error system (7) becomes a linear system, where $K(t)$ as a function of the error vector $E(t)$ is a control function.

By choosing a suitable controller function $K(t)$, the system (7) becomes stable. We choose

$$K(t) = RE(t), \tag{8}$$

where $R$ is an $n \times n$ constant matrix.

The system (7) becomes

$$D^\alpha E(t) = (Q + P + R)E(t). \tag{9}$$

A matrix $R$ is chosen in such a manner that the eigenvalues of $Q + P + R$ are satisfied for $arg(\mu) > \frac{q\pi}{2}$. This appropriate choice will lead to

$$\lim_{t \to \infty} \| E(t) \| = 0, \tag{10}$$

and hence, we achieve HCPS between the considered master system and the slave system.

## 4. Synchronization Phenomena

The fractional order complex Lorenz system is

$$\begin{aligned} D^\alpha u'_1 &= p_1(u'_2 - u'_1) \\ D^\alpha u'_2 &= p_2 u'_1 - u'_2 - u'_1 u'_3 \\ D^\alpha u'_3 &= \frac{1}{2}(\overline{u'_2} u'_2 + u'_1 \overline{u'_2}) - p_3 u'_3, \end{aligned} \tag{11}$$

where $u' = [u'_1, u'_2, u'_3]^T$ is the state variable vector, $u'_1 = u_1 + iu_2$, $u'_2 = u_3 + iu_4$ are complex variables, $u'_3 = u_5$ is the real variable and $p_1, p_2, p_3$ are real constant parameters.

Separating into real and imaginary parts, we have the system (11)

$$
\begin{aligned}
D^\alpha u_1 &= p_1(u_3 - u_1) \\
D^\alpha u_2 &= p_1(u_4 - u_2) \\
D^\alpha u_3 &= p_2 u_1 - u_3 - u_1 u_5 \\
D^\alpha u_4 &= p_2 u_2 - u_4 - u_2 u_5 \\
D^\alpha u_5 &= u_1 u_3 - u_2 u_4 - p_3 u_5.
\end{aligned}
\tag{12}
$$

For the values of parameters as $p_1 = 10, p_2 = 180, p_3 = 1$, initial conditions as $u(0) = [2, 3, 5, 6, 9]^T$ with $\alpha = 0.94$, the system is chaotic.

The fractional order complex T-system is

$$
\begin{aligned}
D^\alpha v_1' &= q_1(v_2' - u_1') \\
D^\alpha v_2' &= (q_2 - q_1) u_1' - q_1 v_1' v_3' \\
D^\alpha v_3' &= \frac{1}{2}(\overline{v_1'} v_2' + v_1' \overline{v_2'}) - q_3 v_3'
\end{aligned}
\tag{13}
$$

where $v' = [v_1', v_2', v_3']^T$ is the state variable vector, $v_1' = v_1 + iv_2, v_2' = v_3 + iv_4$ are complex variables, $v_3' = v_5$ is the real variable and $q_1, q_2, q_3$ are real constant parameters.

Separating real and imaginary parts, we have

$$
\begin{aligned}
D^q v_1 &= q_1(v_3 - v_1) \\
D^q v_2 &= q_1(v_4 - v_2) \\
D^q v_3 &= (q_2 - q_1) v_1 - q_1 v_1 v_5 \\
D^q v_4 &= (q_2 - q_1) v_2 - q_1 v_2 v_5 \\
D^q v_5 &= v_1 v_3 + v_2 v_4 - q_3 v_5
\end{aligned}
\tag{14}
$$

For the values of parameters as $q_1 = 2.1, q_2 = 30, q_3 = 0.6$; initial condition $v(0) = [8, 7, 6, 8, 7]^T$ with $\alpha = 0.94$, the system is chaotic.

We now consider system (12) as the master and system (14) as a slave, the slave system with the control function $\sigma(t) = [\sigma_1, \sigma_2, \sigma_3, \sigma_4, \sigma_5]$ is given as

$$
\begin{aligned}
D^q v_1 &= q_1(v_3 - v_1) + \sigma_1 \\
D^q v_2 &= q_1(v_4 - v_2) + \sigma_2 \\
D^q v_3 &= (q_2 - q_1) v_1 - q_1 v_1 v_5 + \sigma_3 \\
D^q v_4 &= (q_2 - q_1) v_2 - q_1 v_2 v_5 + \sigma_4 \\
D^q v_5 &= v_1 v_3 + v_2 v_4 - q_3 v_5 + \sigma_5.
\end{aligned}
\tag{15}
$$

The synchronization errors can be written as:

$$
e_i' = v_i' - \mu_i u_i',
$$

where $e_1' = e_1 + ie_2, e_2' = e_3 + ie_4, e_3' = e_5$.

On separating real and imaginary parts, we obtain

$$
\begin{aligned}
e_1 &= v_1 - \mu_1 u_1 + \mu_2 u_2 \\
e_2 &= v_2 - \mu_1 u_2 - \mu_2 u_1 \\
e_3 &= v_3 - \mu_3 u_3 + \mu_4 u_4 \\
e_4 &= v_4 - \mu_3 u_4 - \mu_4 u_3 \\
e_5 &= v_5 - \mu_5 u_5.
\end{aligned}
\tag{16}
$$

The error dynamics is obtained as:

$$
\begin{aligned}
D^{\alpha} e_1 &= q_1(v_3 - v_1) - \mu_1 p_1(u_3 - u_1) + \mu_2 p_2(u_4 - u_2) + \sigma_1 \\
D^{\alpha} e_2 &= q_1(v_4 - v_2) - \mu_1 p_2(u_4 - u_2) - \mu_2 p_1(u_3 - u_1) + \sigma_2 \\
D^{\alpha} e_3 &= (q_2 - q_1)v_1 - q_1 v_1 v_5 - \mu_3(p_2 u_1 - u_3 - u_1 u_5) + \mu_4(p_2 u_2 - u_4 - u_2 u_5) + \sigma_3 \\
D^{\alpha} e_4 &= (q_2 - q_1)v_2 - q_1 v_2 v_5 - \mu_3(p_2 u_2 - u_4 - u_2 u_5) + \mu_4(p_2 u_1 - u_3 - u_1 u_5) + \sigma_4 \\
D^{\alpha} e_5 &= v_1 v_3 + v_2 v_4 - q_3 v_5 - \mu_5(u_1 u_3 + u_2 u_4 - p_3 u_5) + \sigma_5.
\end{aligned}
\tag{17}
$$

The control functions are designed appropriately as:

$$
\begin{aligned}
\sigma_1 &= \mu_1 p_1 u_3 - \mu_2 p_1 u_4 - \mu_3 q_1 u_3 + \mu_4 q_1 u_4 + \mu_1 q_1 u_1 - \mu_2 q_1 u_2 - p_1 v_1 + K_1(t) \\
\sigma_2 &= \mu_1 p_1 u_4 + \mu_2 p_1 u_3 - \mu_3 q_1 u_4 - \mu_4 q_1 u_3 + \mu_1 q_1 u_2 + \mu_2 q_1 u_1 - p_1 v_2 + K_2(t) \\
\sigma_3 &= q_1 v_1 v_5 + \mu_3 p_2 u_1 - \mu_3 u_1 u_5 - \mu_4 p_2 u_2 + \mu_4 u_2 u_5 - (q_2 - q_1)\mu_1 u_1 + (q_2 - q_1)\mu_2 u_2 - v_3 + K_3(t) \\
\sigma_4 &= q_1 v_2 v_5 + \mu_3 p_2 u_2 - \mu_3 u_2 u_5 + \mu_4 p_2 u_1 - \mu_4 u_1 u_5 - (q_2 - q_1)\mu_1 u_2 + (q_2 - q_1)\mu_2 u_1 - v_4 + K_4(t) \\
\sigma_5 &= -v_1 v_3 - v_2 v_4 + q_3 \mu_5 u_5 + \mu_5 u_1 u_3 + \mu_5 u_2 u_4 + p_3 v_5 + K_5(t).
\end{aligned}
\tag{18}
$$

Thus, the resulting error system is given by

$$
\begin{aligned}
D^{\alpha} e_1 &= q_1 e_3 - q_1 e_1 - p_1 e_1 + K_1(t) \\
D^{\alpha} e_2 &= q_1 e_4 - q_1 e_2 - p_1 e_2 + K_2(t) \\
D^{\alpha} e_3 &= (q_2 - q_1)e_1 - e_3 + K_3(t) \\
D^{\alpha} e_4 &= (q_2 - q_1)e_2 - e_4 + K_4(t) \\
D^{\alpha} e_5 &= -q_3 e_5 - p_3 e_5 + K_5(t).
\end{aligned}
\tag{19}
$$

As $K(t) = Re(t)$, where $R$ is $5 \times 5$ constant matrix and $e(t) = [e_1(t), e_2(t), e_3(t), e_4(t), e_5(t)]^T$. Here, we choose matrix $R$ as follows:

$$
\begin{bmatrix}
q_1 + p_1 - r & 0 & -q_1 & 0 & 0 \\
0 & q_1 + p_1 - r & 0 & -q_1 & 0 \\
-(q_2 - q_1) & 0 & 1 - r & 0 & 0 \\
0 & -(q_2 - q_1) & 0 & 1 - r & 0 \\
0 & 0 & 0 & 0 & q_3 + p_3 - r
\end{bmatrix}.
$$

Then, the eigenvalues of the linear system (19) are $(-r, -r, -r, -r, -r)$. For simplicity, we take $r = 1$, then the resultant error system for this particular choice becomes

$$
D^{\alpha} e_i(t) = -e_i(t), \quad i = 1, 2, 3, 4, 5
\tag{20}
$$

## 5. Numerical Simulation and Discussion

Numerical simulation is performed to explore and validate the effectiveness and superiority of the suggested HCPS scheme between fractional order complex chaotic Lorenz system (master) and fractional order complex chaotic T-system (slave). We here simply utilize the 4th-order Runge–Kutta algorithm along with the Oustaloup's technique to approximate the fractional derivatives. In computation, consider $\alpha = 0.94$ with a step size of 0.005. The initial conditions for the master system and the slave system are [2,3,5,6,9] and [8,7,6,8,7], respectively. Thus, according to the considered error system, the initial conditions are $(e_1, e_2, e_3, e_4) = [6.3, 3.8, 8, 16.5, -3.8]$. The complex scaling matrix is $\mu = [1 + 0.1i, -1 - 0.05i, 1.2]$. The chaotic behavior of the fractional order complex Lorenz system and T-system has been shown in Figures 1a–d and 2a–d, which represent the phase diagrams of the considered systems. It is clearly shown that Figure 3a–c displays the state trajectories of the master system and the controlled slave system behaving alike. Additionally, Figure 4a,b represents that the synchronization error goes to zero as time tends to infinity. By choosing different complex scaling matrices, we can synchronize the given

systems up to the desired result. Thus, the considered HCPS strategy between dissimilar master and slave systems is validated computationally using MATLAB environment.

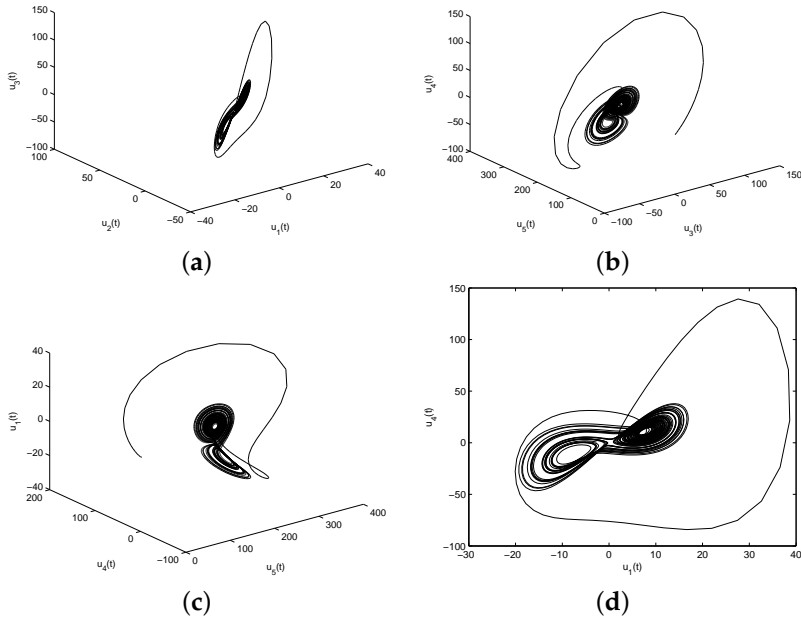

**Figure 1.** Phase diagrams of fractional order complex Lorenz system (11) and (12) at $q = 0.95$; (**a**) between $u_1(t) - u_2(t) - u_3(t)$ space; (**b**) between $u_3(t) - u_5(t) - u_4(t)$ space; (**c**) between $u_5(t) - u_4(t) - u_1(t)$ space; (**d**) between $u_1(t) - u_4(t)$ plane.

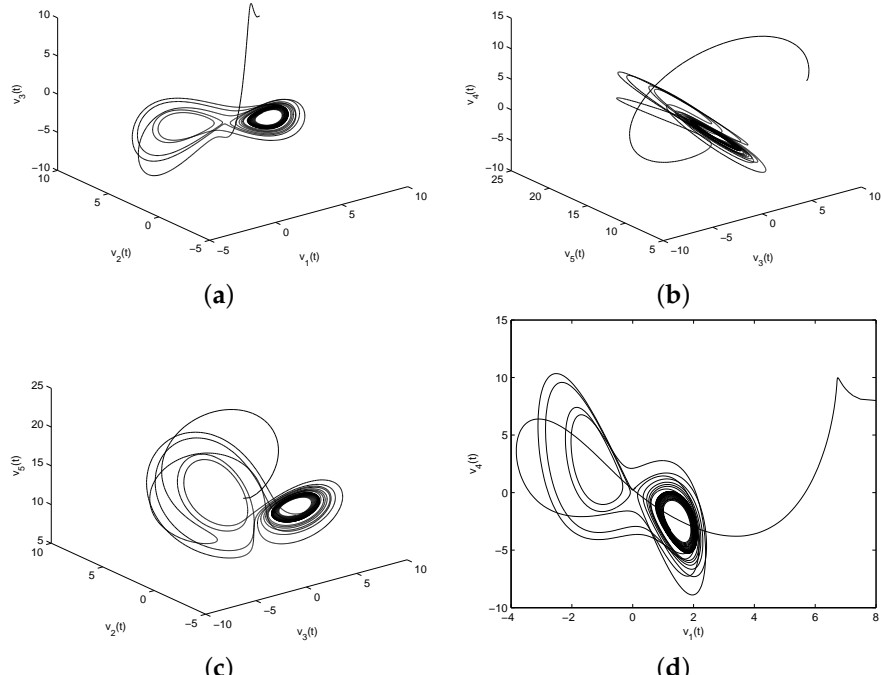

**Figure 2.** Phase diagrams of slave fractional order complex chaotic T-system for (13) and (14) at $q = 0.95$; (**a**) between $v_1(t) - v_2(t) - v_3(t)$ space; (**b**) between $v_3(t) - v_5(t) - v_4(t)$ space; (**c**) between $v_3(t) - v_2(t) - v_5(t)$ space; (**d**) between $v_1(t) - v_4(t)$ plane.

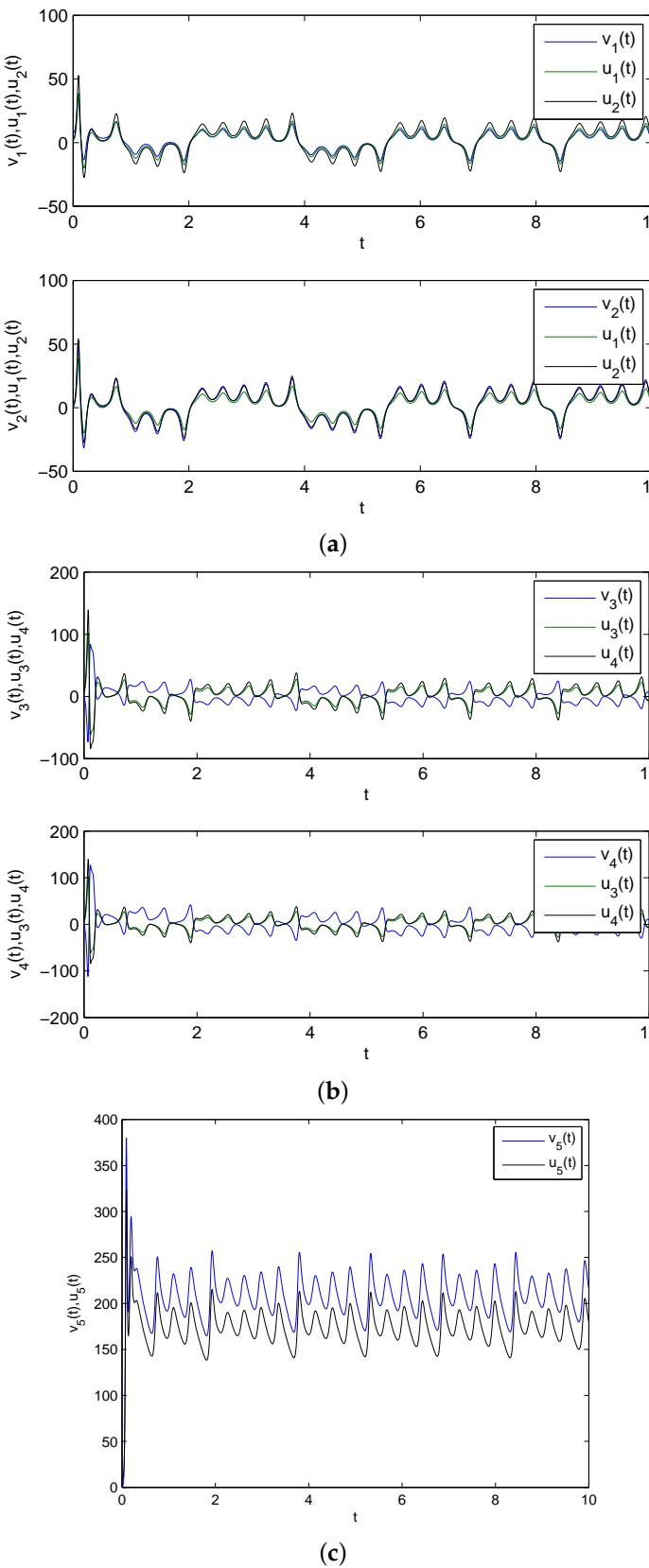

**Figure 3.** State trajectories of master fractional order complex chaotic system and controlled slave fractional order complex chaotic system for (11)–(14) at $q = 0.95$; (**a**) $v_1(t) - u_1(t) - u_2(t)$ and $v_2(t) - u_1(t) - u_2(t)$; (**b**) $v_3(t) - u_3(t) - u_4(t)$ and $v_4(t) - u_3(t) - u_4(t)$; (**c**) $v_5(t) - u_5(t)$.

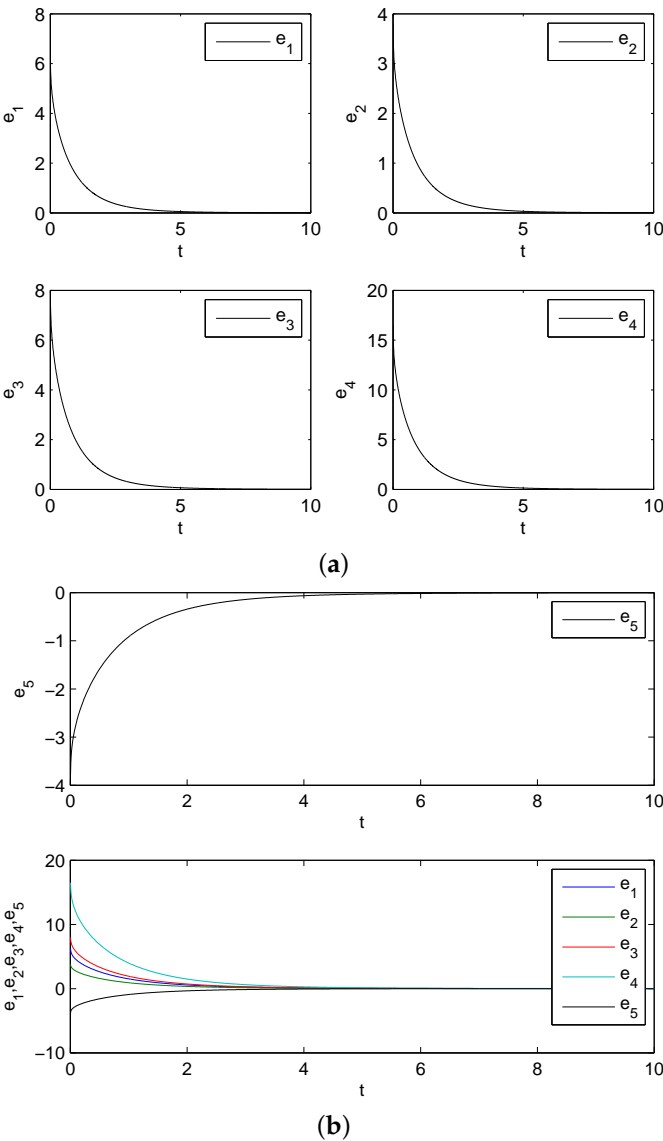

**Figure 4.** Synchronization error between master system and slave system at $q = 0.95$ with controllers;
(**a**) $(t, e_1)$; $(t, e_2)$; $(t, e_3)$ and $(t, e_4)$; (**b**) $(t, e_5)$ and simultaneous plot $(t, e_1, e_2, e_3, e_4, e_5)$.

## 6. Conclusions

In this research work, a hybrid complex projective synchronization (HCPS) methodology has been investigated among dissimilar two fractional order chaotic complex systems via an active control strategy. Appropriate active nonlinear control input has been designed because of the drive-response/master–slave configuration and Lyapunov's stability analysis (LSA). The error dynamical system converges asymptotically to zero by utilizing a proper and simplified nonlinear active control input. The efficacy and superiority of the analytical outcomes are validated by executing simulation through the MATLAB environment. Significantly, both the analytic work and the numerical effects are in excellent conformity. We noted that our investigated HCPS method is primary yet theoretically precise. Further, this scheme will serve as a prominent task in enhancing security in communication and encrypting images with numerous applications in biological, social, and physical nonlinear dynamic systems. Moreover, we realize that the suggested HCPS methodology may be generalized via other control schemes.

**Author Contributions:** Conceptualization, M.S., H.C. and S.K.; methodology, M.S. and H.C.; validation, S.K.; formal analysis, H.C.; investigation, H.C.; writing—original draft, H.C. and S.K.; writing—review and editing, M.S.; visualization, H.C.; funding acquisition, M.S. All authors have read and agreed to the published version of the manuscript.

**Funding:** This research was funded by Deanship of Scientific Research, Qassim University grant number 10163-qec-2020-1-3-I.

**Data Availability Statement:** Not applicable.

**Acknowledgments:** The authors gratefully acknowledge Qassim University, represented by the Deanship of Scientific Research, on the financial support for this research under the number 10163-qec-2020-1-3-I during the academic year 1441 AH/2020 AD.

**Conflicts of Interest:** The authors declare no conflict of interest.

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
