# Peer review of "Chaos Controllability in Non-Identical Complex Fractional Order Chaotic Systems via Active Complex Synchronization Technique"

_axioms, doi:10.3390/axioms12060530_

Round 1

Reviewer 1 Report

In this paper, the authors develop a methodology for complex synchronization schemes in general fractional order chaotic systems. A design of master-slave systems is done via an appropriate control function in the slave system in such a way that the controlled slave system (equation (2) in the text) asymptotically converges to the master system (expression (1) in the article). A suitable matrix is also chosen in order the error  goes to zero when time increases indefinitely. The paper finishes with some numerical simulations using the method described, and a section of conclusions.

I think the paper is well written and is interesting for the readers of the journal. Some formal aspects could be improved, as for instance:

-There is an excessive use of abbreviations in the first part of the paper.

-The notation u'i is a bit misleading in this context. It should be explained or changed.

-There are very few typos: In line 158 write "active", Lorenz must be capitalized in bibliography (see [27] and [30]).

These considerations are more suggestions than corrections, thus the article could be published in its current form after a revision by the editorial managing office.

The English writing is correct

Author Response

Thanks for providing comments and suggestions. Please see attached file for author's responses.

Reviewer 2 Report

In this paper, the authors propose the synchronisation of non identical complex fractional order chaotic systems via an active complex technique. The obvious interest of complex chaotic systems is to increase the dimension of the resulting set of differential equations and consequently to improve encryption and security of communications. The graphs of figure 4 exhibit the interest of the proposed synchronisation methodology.

However, I have three basic comments.

The first one is related to the numerical simulation procedure. A fundamental requirement of science is that any experiment can be reproduced by peers. The authors tell that Matlab simulation has been performed using the Runge Kutta 4 technique which is considered as a standard for the simulation of integer order differential systems. However, a fractional order system is very different from an integer order one. Thus, it is absolutely necessary to tell how the fractional order feature of the chaotic systems is took into account with RK4. I suppose that the authors have transformed the fractional equations into integer ones using an approximation technique. This approximation is not straightforward neither trivial. Moreover, it highly conditions the experimental curves and the reproducibility of the results. Thus, I consider that it is mandatory to provide this essential information in the paper.

The second comment is related to the reference to Lyapunov stability. The procedure described in the paper refers to the linear equations 8 and 9. Their application to the complex Lorenz chaotic systems is apparently based on classical linear system theory, without reference to Lyapunov stability. In fact, I think that all the proposed methodology is based on Lyapunov theory. Thus I think it would be necessary to express explicitly where the Lyapunov method is used. Moreover, since global synchronisation is based on classical synchronisation and anti synchronisation, as exhibited by the graphs of figure 4, I think it would be necessary to provide more information on the different a priori choices.

The third comment is related to initial conditions which are a fundamental feature of chaotic systems. This comment is not particularly specific to this paper since all researchers think that Caputo initial conditions are the same as integer order derivative initial conditions. Fractional order derivatives depend on all the past behaviour of the system, not only on the last value x(t0). This means that the real interest of fractional chaotic systems is that their behaviour depends on more complex initial conditions, which cannot be summarized by the pseudo state x(t0). This is a fundamental contradiction linked to the use of the Caputo derivative. At least, it is necessary to insist on the fact that x(t0) is only a pseudo initial condition.

none

Author Response

(The authors gave the same response as above.)

Author Response

(The authors gave the same response as above.)

Round 2

Reviewer 2 Report

I am not satisfied by the responses to my comments, and more particularly to my comment 1. I asked the authors to tell the future readers to explain how the chaotic system is simulated. In their response, they say the response is in reference [35] !

This is not a response. The readers of your paper need to know the technique used by the authors, and it requires a real explanation which cannot be reduced to a simple reference.

Up to my knowledge, there are two main techniques to simulate fractional systems: the Grünwald technique and the approximation of the fractional derivative by the Oustaloup's technique. If you use RK4 algorithm, this means that you refer to the second technique.

Thus, I ask you to provide a complete information in section 5, with the essential details in order that anyone else can reproduce your results.

For comment 2, I agree with the authors, but their answer would have been to be placed in the corresponding section.

For comment 1, you only have introduced the word "pseudo" in section 4 page 6. In my opinion, once again, this is not sufficient. In fact, it would be necessary to introduce more convincing remarks in section 2 and 3, related to the new reference [36].

no comment

Author Response

Thanks for providing again chance to revise the manuscript. Sorry for inconvenience since we could not understand fully all comments and suggestions of reviewer and these were not updated according to the reviewer point of view. We are again trying to incorporate all possible suggestions. The changes made after Revision 2 in the text are highlighted in Magenta colour and the changes made after Revision 1 in the text are highlighted in Green colour. Please see below again responses which are enumerated herein by providing explanation:

Comment 1: The first one is related to the numerical simulation procedure. A fundamental requirement of science is that any experiment can be reproduced by peers. The authors tell that MATLAB simulation has been performed using the Runge-Kutta 4 technique which is considered as a standard for the simulation of integer order differential systems. However, a fractional order system is very different from an integer order one. Thus, it is absolutely necessary to tell how the fractional order feature of the chaotic systems is took into account with RK4. I suppose that the authors have transformed the fractional equations into integer ones using an approximation technique. This approximation is not straightforward neither trivial. Moreover, it highly conditions the experimental curves and the reproducibility of the results. Thus, I consider that it is mandatory to provide this essential information in the paper.

Answer: First of all, authors appreciate the esteemed reviewer for his valuable comments. Fractional order dynamics is not much different from integer order dynamics. Fractional order dynamics satisfies the integer order dynamics whenever we take the order of fractional dynamics as an integer.  We are not approximating the fractional order into integer order for any numerical simulation or MATLAB programming. In fact, we find the approximate solutions of simultaneous nonlinear equations using fourth-order Runge-Kutta method in which we use the Oustaloup's technique to approximate the fractional derivatives (Page: 12, Section-5). It (fractional dynamics) has its own methodologies through that we find the simulation and write the MATLAB programs.

Comment 2: The second comment is related to the reference to Lyapunov stability. The procedure described in the paper refers to the linear equations 8 and 9. Their application to the complex Lorenz chaotic systems is apparently based on classical linear system theory, without reference to Lyapunov stability. In fact, I think that all the proposed methodology is based on Lyapunov theory. Thus I think it would be necessary to express explicitly where the Lyapunov method is used. Moreover, since global synchronization is based on classical synchronization and anti-synchronization, as exhibited by the graphs of figure 4, I think it would be necessary to provide more information on the different a priori choices.

Answer: In the considered master-slave configuration, the states of the slave system are evolving over the period of time which is guided by active controller and this active controller is obtained by the error dynamics equation using Hybrid Complex Projective Synchronization (HCPS) scheme. This error dynamics is results of the both master and slave chaotic systems. As a consequence, we have the required error dynamics. Further, this error dynamics has been used in order to show that the derivative of the designed Lyapunov function is negative definite function. Consequently, the considered error dynamics achieve global and asymptotical stability with the evolution of time. As suggested, we have added the above mentioned facts in our revised manuscript (Page: 3, Section-1, Para-3).                                                                                         

Comment 3: The third comment is related to initial conditions which are a fundamental feature of chaotic systems. This comment is not particularly specific to this paper since all researchers think that Caputo initial conditions are the same as integer order derivative initial conditions. Fractional order derivatives depend on all the past behaviour of the system, not only on the last value x(t0). This means that the real interest of fractional chaotic systems is that their behaviour depends on more complex initial conditions, which cannot be summarized by the pseudo state x(t0). This is a fundamental contradiction linked to the use of the Caputo derivative. At least, it is necessary to insist on the fact that x(t0) is only a pseudo initial condition.

 Answer: We have agreed with the comments that the Caputo initial condition is as same as in that of the integer order derivatives. Moreover, the main advantage of the Caputo approach is that the initial conditions defined for Caputo differential fractional differential equations and the initial conditions defined for integer order differential equations are the same. Therefore, Caputo fractional derivative operator is preferred instead of Riemann-Liouville fractional derivative operator. Another important difference between the Riemann-Liouville derivative definition and the Caputo derivative definition is that although the Caputo derivative of the constant is zero, the Riemann-Liouville fractional derivative of the constant is not equal to zero for a finite value of an arbitrary real α. As suggested, we have provided this information in our revised manuscript (Page: 2-3, Section-1). So, we have deleted "pseudo" from initial conditions. In addition, fractional order dynamical system’s behaviors has its own nature and phase portraits. But, when we take the order of fractional dynamics as an integer, it behave as integer order dynamical systems. Researchers have interested in that when the order of fractional dynamics is not integer then what is the nature and behaviors of fractional dynamics in the form of initial conditions, methodologies and phase portraits.  

We are again grateful to the referee' for their valuable comments and suggestions. We have tried our best to improve the quality of the manuscript. However, these changes will not influence the content and framework of the paper. We hope that the correction will meet with approval. 

Reviewer 3 Report

No comments.

Author Response

Thanks for providing recommendation to publish in Axioms.

Round 3

Reviewer 2 Report

I agree with the improvements of last version of the revised proposal